# An Ice-Binding Protein from an Antarctic Ascomycete Is Fine-Tuned to Bind to Specific Water Molecules Located in the Ice Prism Planes

**DOI:** 10.3390/biom10050759

**Published:** 2020-05-13

**Authors:** Akari Yamauchi, Tatsuya Arai, Hidemasa Kondo, Yuji C. Sasaki, Sakae Tsuda

**Affiliations:** 1Graduate School of Life Science, Hokkaido University, Sapporo 060-0810, Japan; h.kondo@aist.go.jp; 2Graduate School of Frontier Sciences, The University of Tokyo, Kashiwa 277-8561, Japan; t.arai@edu.k.u-tokyo.ac.jp (T.A.); ycsasaki@edu.k.u-tokyo.ac.jp (Y.C.S.); 3Bioproduction Research Institute, National Institute of Advanced Industrial Science and Technology (AIST), Sapporo 062-8517, Japan; 4OPERANDO Open Innovation Laboratory, National Institute of Advanced Industrial Science and Technology (AIST), Tsukuba 305-8563, Japan

**Keywords:** ice-binding protein, antifreeze protein, ascomycete, thermal hysteresis, fluorescence-based ice plane affinity, polygonal waters

## Abstract

Many microbes that survive in cold environments are known to secrete ice-binding proteins (IBPs). The structure–function relationship of these proteins remains unclear. A microbial IBP denoted *Anp*IBP was recently isolated from a cold-adapted fungus, *Antarctomyces psychrotrophicus*. The present study identified an orbital illumination (prism ring) on a globular single ice crystal when soaked in a solution of fluorescent *Anp*IBP, suggesting that *Anp*IBP binds to specific water molecules located in the ice prism planes. In order to examine this unique ice-binding mechanism, we carried out X-ray structural analysis and mutational experiments. It appeared that *Anp*IBP is made of 6-ladder β-helices with a triangular cross section that accompanies an “ice-like” water network on the ice-binding site. The network, however, does not exist in a defective mutant. *Anp*IBP has a row of four unique hollows on the IBS, where the distance between the hollows (14.7 Å) is complementary to the oxygen atom spacing of the prism ring. These results suggest the structure of *Anp*IBP is fine-tuned to merge with the ice–water interface of an ice crystal through its polygonal water network and is then bound to a specific set of water molecules constructing the prism ring to effectively halt the growth of ice.

## 1. Introduction

Proteins bind to their ligands to perform specific functions. Hydration water molecules in a protein play a significant role in ligand binding and also have roles in dissolving proteins in water and maintaining their tertiary structures [1,2]. Some hydration water molecules located on an ice-binding protein (IBP) are known to be arranged polygonally, similar to the hexagonally latticed waters of ice crystals formed under atmospheric pressure [3,4]. The presence of regularly arrayed troughs, which trap the hydration water molecules at regular intervals, has also been reported in some IBPs [5,6,7,8,9]. These water molecules are arranged in an ice-like geometry and are thus called “ice-like waters”. A role for these molecules in anchoring the host protein to ice crystal surfaces has been postulated. 

The IBPs isolated from cold-adapted organisms such as fish, insects, plants, and microorganisms [10] are known to inhibit ice growth through binding to the water molecules constructing the crystal surface. Such ice, a ligand of IBP, is a solid state of water in which the water molecules are systematically organized into a hexagonal arrangement. This arrangement is the principle feature of the single ice crystal created at one atom defined by three equivalent *a*-axes (*a*_1_–*a*_3_) perpendicular to the *c*-axis [11]. The hexagonal ice (denoted I_h_) has six primary prism planes and six secondary prism planes along the *c*-axis. These prism planes lie parallel and perpendicular to the *a*-axes, respectively. The basal plane lies perpendicular to the *c*-axis and parallel to the *a*-axis. Other inclined planes between the basal and prism planes are defined as pyramidal planes. It has been demonstrated that the growth speed of the prism planes toward the *a*-axis is approximately 100 times faster than that of the basal planes toward the *c*-axis [12]. A snowflake is an example of a hexagonal single ice crystal, though its preparation is very difficult. An alternative handling technique for single ice crystals was recently developed [13], in which a single ice crystal hemisphere 3–4 cm in diameter is created by utilizing a plastic pipe and a cooling bath. We can determine the *a*- and *c*-axes of this golf-ball-size single ice crystal using a polarizer, where its earth-axis corresponds to the *c*-axis of I_h_. When such an ice hemisphere is mounted on a frosty probe with the desired orientation, and soaked in a solution of fluorescence-labeled IBP, a pattern appears on the globular ice surface under ultraviolet (UV) light. This is called the pattern of fluorescence-based ice plane affinity (FIPA) of an IBP species, as presented in [14]. Variations of the FIPA pattern have recently been reported [15]. From these data, one can correlate the ice planes constructing the hexagonal single ice crystal and the FIPA patterns created on the globular single ice crystal. For example, the FIPA pattern observed at the polar area of a globular single ice crystal implies the basal-plane binding of an IBP.

In addition to the FIPA pattern, IBP’s ice-binding ability can be detected by the observation of morphological change in an ice crystal and thermal hysteresis (TH) activity. Both observations require a photomicroscope system equipped with a cooling stage, whose temperature needs to be lowered to deep freezing temperatures, such as −40 °C. IBP changes the photomicroscope image of a single ice crystal into a hexagonal bipyramid, a hexagonal trapezohedron [14,15], or a lemon-like shape [6], depending upon the type and number of ice planes growth terminated by an IBP species [5]. TH is the difference between the melting and freezing temperatures (*T*_m_ and *T*_f_) of a single ice crystal prepared in an IBP solution; TH = |*T*_m_ − *T*_f_| [14,16,17,18]. The two temperatures are equal for normal solutions, and the presence of IBPs decreases *T*_f_ and slightly raises *T*_m_ to generate the TH, which can be evaluated with a photomicroscope system as described earlier [19]. IBP from fish and plants generally exhibit 0.5–1.5 °C of TH activity at millimolar concentrations, and they are called “moderately active IBP”. In contrast, insect and microbial IBPs that exhibit 2–6 °C of TH activity even at micromolar concentrations are called “hyperactive IBPs” [6]. 

IBPs have been used to elucidate the relationship between hydration water molecules and the ice-binding function. IBPs have one or two ice-binding sites (IBSs), which consist of many hydrophobic residues making a flat surface in which several polar residues are inserted at regular intervals or dispersively [5]. For several IBP species, “caged” hydration water molecules were observed on the hydrophobic IBS with some hydrogen bonding to the polar residues. The caged water molecules are not hexagonally organized but exhibit a pentagonal arrangement and therefore are called “ice-like” water molecules [4,20,21,22]. For example, IBP from *Marinomonas primoryensis* (*Mp*IBP) has pentagonally arranged water molecules on its IBS [4]. These molecules exist in the vicinity of CH_3_ groups of Thr side chains and are stabilized on the IBS through hydrogen bonding to the main-chain nitrogen and side-chain hydroxyl groups of Thr, as well as to the side-chain oxygen atom of Asp. The ice-like water molecules are thought to play a significant role in facilitating IBP binding to ice crystals and are known as anchored clathrate water (ACW). Polygonally arranged water molecules have been observed for moderately active type II and III IBPs [20,21], as well as hyperactive type I IBP (Maxi) [22,23], but not for insect IBPs [5,6,7,8,9]. Instead, surface water molecules are trapped in regularly arrayed troughs on insect IBPs, which are made of two rows of Thr residues, and the spacing of the oxygen atom of the trapped water molecules matches the distance between water molecules in both basal and prism planes of a single ice crystal. Such troughs are created by the repetitive nature of the amino acid sequence of IBPs from *Tenebrio molitor* (*Tm*IBP) and *Dendroides canadensis* (*D*IBP), which consists of a 12-residue consensus sequence CTxSxxCxxAxT repeated tandemly, where C, T, S, and A are Cys, Thr, Ser, and Ala, respectively, and x represents any residue [9,24]. *Anp*IBP does not have any tandem repeat sequences [25], and it is not clear what structure it forms or how it binds to ice crystals.

IBPs from microorganisms including bacteria [4], fungi [26], diatoms [27], and copepods [28] share a common structure, denoted the domain of unknown function (DUF3494) [29]. This domain is constructed of approximately 230 amino acids (~25 kDa) and speculated to be widespread, having been passed many times between prokaryotic and eukaryotic microorganisms by horizontal gene transfer [25,28]. The DUF3494 domain is characterized by a uniquely formed β-helical structure, which is also assumed to exist in *Anp*IBP. IBP from the Antarctic bacterium *Colwellia sp.* SLW05 (*Col*IBP) and that from Arctic yeast (*Le*IBP) are both contain the DUF3494 domain. The former exhibits high TH activity (~4 °C) at 140 µM [30], whereas the activity of *Le*IBP is much lower (0.35 °C), even at 370 µM [31]. The *Anp*IBP exhibited TH activity (~0.7 °C) at 300 µM, which was the same level as that of the moderately active fish IBP. An ability to modify ice crystals into a lemon shape is regarded as a typical sign of high TH activity. *Anp*IBP was reported to generate such a lemon-shaped crystal, although it is only moderately active [25]. The present study used the FIPA experiments and X-ray crystal structure determination of a recombinant protein of *Anp*IBP and a mutant version for which an extraordinary ice-binding property has been suggested. Here we present new structural data for an IBP identified from a cold-tolerant fungus, *Antarctomyces psychrotrophicus* (*Anp*IBP). *Anp*IBP showed a unique ability to bind to the equator region of a single ice crystal when we prepared it in a globular form, a phenomenon that we called a “prism ring”. 

## 2. Materials and Methods

### 2.1. Expression and Purification of Recombinant AnpIBP and Its Mutants

Wild-type *Anp*IBP and a mutant version were prepared as described previously, using *Pichia pastoris* as the host organism [25]. Briefly, a DNA encoding His-tag (His × 6) plus the Ala1-Val216 sequence of *Anp*IBP was inserted downstream of an α-factor signal sequence in the expression vector pPICZα. The vector was transformed into *P.*
*pastoris* after linearization by a restriction enzyme. The transformant was precultured in buffered glycerol complex medium, followed by 96–120 h incubation in methanol-containing medium, to express *Anp*IBP. The culture was then centrifuged to collect the medium containing *Anp*IBP. The protein was purified by a series of ion-exchange chromatography passes (High S and High Q, Bio-Rad, Hercules, CA, USA) and further purified by size-exclusion chromatography (Superdex 200, GE Healthcare, Chicago, IL, USA). The mutant proteins were prepared using the same procedure. The purity of these samples was confirmed by sodium dodecyl sulfate-polyacrylamide gel electrophoresis (SDS-PAGE). We have previously reported that *A.*
*psychrotrophicus* secretes glycosylated *Anp*IBP at the Asn55 position [25]. The *Anp*IBP under investigation here, expressed in *P. pastoris*, was also thought to be glycosylated, because it showed multiple or smear bands in SDS-PAGE. We found that the substitution of Asn55 with aspartic acid produced a non-glycosylated form of *Anp*IBP, and the N55D mutant of *Anp*IBP is defined as *Anp*IBP hereafter. Further amino acid replacements were made in this *Anp*IBP. The S153 of *Anp*IBP was replaced with Tyr, for example, and is called the S153Y mutant in this article. These samples were dissolved in water or 20 mM Tris–HCl at pH 8.0, and concentrated by ultrafiltration (MW cutoff = 3 kDa) (Merck Millipore, Burlington, MA, USA). Their final concentration was determined using UV absorbance at 280 nm with a coefficient calculated on the basis of the amino acid sequence.

### 2.2. Thermal Hysteresis Measurements and Ice Crystal Morphology 

The TH activity of *Anp*IBP and its mutants was measured by observation of a single ice crystal prepared in an IBP solution, using a photomicroscope equipped with a temperature-controlled stage and a Charge Coupled Device (CCD) camera, as described previously [19]. A 1-µL sample solution in a glass capillary tube was flash frozen to approximately −25 °C and slowly warmed to near 0 °C to observe a single ice crystal. For this crystal, we initially measured the melting point (*T*_m_) while increasing the stage temperature. We then prepared another single ice crystal, and lowered the temperature with a cooling rate of 0.1 °C/min. An ice crystal does not start to grow when the temperature is higher than the lower limit of TH, and it shows bursting growth when this limit is exceeded, which is determined to be the freezing point (*T*_f_). The difference between the two temperatures (*T*_m_ − *T*_f_) is defined as the TH value [14,16,17,18]. The TH was measured at least three times, and an averaged value was plotted at concentrations between 30 and 500 µM.

### 2.3. FIPA Analysis

In order to identify the IBP-bound plane of a single ice crystal, the FIPA pattern was observed for our samples using the procedure described in [14]. A single ice crystal was prepared in a polyvinyl chloride (PVC) pipe (Ø = 30 mm and height = 30 mm) with a tiny notch (1 mm × 2 mm) placed on the aluminum tray. Degassed water was poured into the tray to 5 mm in depth and cooled to 0.5 °C. Ice growth was initiated by adding a small piece of ice to the supercooled water, which was propagated into the PVC pipe though the notch, acting as a seed to generate a single ice crystal. When the inside of the pipe started freezing, degassed water was carefully added into the PVC pipe. After further incubation at −1.8 °C for 3 days, a single cylindrical shaped ice crystal was created in the pipe. The crystallographic *c*- and *a*_1_–*a*_3_-axes of this cylindrical ice crystal were checked using polarizers. The crystal was modified into a hemispherical shape (Ø = 30 mm) with a heated stick and was mounted on a chilled copper rod with a desired orientation. The single ice crystal hemisphere mounted on the rod was soaked in a 40-mL solution of fluorescence-labeled IBP so as to face down the curved front of the crystal. The fluorescence dye used was tetra-methyl-rhodamine (5 (6)-TAMRA-X, SE: Thermo Fisher Scientific, Waltham, MA, USA). The single ice hemisphere was held in the solution for 4–5 h at −5 °C. When it became golf ball sized (Ø = 5 cm), it was detached from the copper rod to observe the FIPA pattern under UV light in a cold room (−1 °C).

### 2.4. Crystallization and X-Ray Structure Determination of AnpIBP and Its Mutants

The *Anp*IBP and its S153Y mutant prepared without His-tag were crystallized using the hanging drop vapor diffusion method [32]. A 1-µL solution of each solution, with the concentration adjusted to 20 mg/mL was mixed with an equal volume of a crystallization solution at 20 °C. Large crystals of wild-type *Anp*IBP were obtained when they were dissolved with 0.1 M HEPES-NaOH pH 8.6, 0.95 M ammonium sulfate, and 0.1 M lithium sulfate. The S153Y mutant was crystallized with 0.1 M Bis-Tris pH 7.0 and 4.0 M ammonium acetate. The experiments using synchrotron radiation were performed with the approval of the Photon Factory Program Advisory Committee (proposal number 2018G105). Diffraction data for *Anp*IBP and the S153Y mutant were collected at the beam line BL-17A [33] at the Photon Factory KEK, Japan, and were processed using XDS [34] and the CCP4 [35] program suit. The crystal structure of *Anp*IBP was determined using the molecular replacement method using Phenix [36] with a coordinate file of *Col*IBP (PDB ID: 3WP9) [30] as a search model. In this model, a 21-residue segment between Pro38 and Leu74 of the original *Col*IBP coordinate was removed, because the corresponding segment does not exist in *Anp*IBP. The initial model structure of *Anp*IBP was further refined using the software packages Phenix, Refmac5 [37], and Coot [38]. The structure of the S153Y mutant was also determined using the molecular replacement method with the coordinate file of *Anp*IBP. The statistics for data collection and refinement [39] are summarized in Appendix A. The crystal packings of wild-type *Anp*IBP and S153Y mutants are shown in Appendix A. The coordinates were deposited in the Protein Data Bank (PDB) under ID 7BWX for *Anp*IBP and 7BWY for the S153Y mutant.

## 3. Results and Discussion

### 3.1. FIPA of AnpIBP Showed a Novel Prism-Ring Pattern

Figure 1A,B shows the FIPA patterns observed for *Anp*IBP, for which illustrated interpretations are presented in Figure 1C,D, respectively. The contiguous illumination ring was observed at the rim of the hemisphere, when it was soaked into the IBP solution to direct its *c*-axis downward (Figure 1A,C). On a different ice hemisphere whose *a*_3_-axis was directed downward, the fluorescence from ice-bound *Anp*IBP was observed in the area between the equator and a middle latitude but not in the polar region (Figure 1B,D). The ring area is centered at the equator, implying that it is made of the water molecules that participate in both the primary prism and the secondary prism planes. For simplicity, we named this orbital FIPA pattern (Figure 1A) a prism ring; *Anp*IBP binds to the water molecules constructing the prism ring. Note that a faint yellow circle observed at the center of hemisphere is an artifact arising from the hole left by the metal rod used to hold and cool the ice crystal [14].

It has been shown that the FIPA pattern is sometimes changed with increases in the concentration of IBP [40,41]. For example, the pattern was observed in a limited area for a 0.01-mg/mL solution of fish type I IBP from barfin plaice, and it was overcast by the entire illumination when the protein concentration was increased to 0.1 mg/mL. This observation was interpreted as indicating that this IBP binds only to the pyramidal plane at 0.01 mg/mL, whereas it covers the whole ice crystal, including the basal plane, when its concentration is 10 times higher [40]. A similar change was observed for type II IBP from longsnout poacher (*Lp*IBP) [41]. We performed a concentration dependence experiment for *Anp*IBP between 0.01 and 0.15 mg/mL. The FIPA pattern did not change with increasing concentration; *Anp*IBP binds to the prism ring but not to the basal plane, even at the highest concentration (data not shown). *Anp*IBP is a member of the microbial IBPs containing the DUF3494 domain. The FIPA pattern reported for the known DUF3494-characterized IBPs, such as *Col*IBP [30], *Tis*IBP [42], *Sf*IBP [43], and *Ff*IBP [44], showed that they are all capable of binding to the basal plane. This observation implies that *Anp*IBP is exceptional, and its manner of binding to the prism ring is controlled by a unique structural property.

### 3.2. AnpIBP Folds as an Irregular β-Helical Structure

To clarify the manner of ice binding, the X-ray crystal structure of *Anp*IBP was determined at 1.9 Å resolution using a molecular replacement method utilizing *Col*IBP (PDB: 3WP9) as a structural template [30]. An asymmetric unit contains six *Anp*IBP molecules (Appendix A). The final model of *Anp*IBP shows the structural coordinates of 203 out of 216 amino acid residues. Statistics of the data collection and the parameter values used in the refinement process are summarized in Appendix A. The structure of *Anp*IBP consists of a 45-Å-long β-helical fold with a triangular cross section (Figure 2A). The main constituents are 6-ladder β-helical loops, which are supported by a long α-helix located along the β-helical axis. The β1 loop following the N-terminal segment (blue) was aligned next to the β6 loop following the C-terminus (red), and they are assembled in a “head-to-tail” manner. As a consequence, the six β-helical loops are irregularly assembled in the order β1–β6–β5–β4–β3–β2.

Such an irregularly ordered β-helical arrangement has been found in other microbial IBPs containing the DUF3494 domain, such as *Col*IBP [30], *Tis*IBP [42], and *Efc*IBP [45]. The β-helical loops of *Anp*IBP are different in length: β1, G12–G29 (18); β6, A184–Q201 (18); β5, N166–G183 (18); β4, N145–G165 (21); β3, S117–A144 (28); and β2, A93–A116 (23). The lengths of the β2–, β3–, and β4–loops are 3- to 10-residues longer than those of the others (β1–, β5–, and β6–loops). The three surfaces of the triangular β-helical domain are named the A-, B-, and C-faces (Figure 2B). The A-face is partially covered by the long α-helix, whereas the other two surfaces are exposed to solvent. As with other microbial IBPs [26,46], there are two hydrophobic cores in *Anp*IBP. One comprises the central area of the β-helical domain, which is contributed by aliphatic residues originated from the β1–β6–β5–β4 region and aromatic residues from the β2–β3–β4 region. A segment from D30 to I52 stacked on the β1-loop does not satisfy the property of β-folds and was assigned as a loop region, protruding aliphatic side chains toward the inner core of the β-helical domain and acting as a capping structure to stabilize the overall structure. Another hydrophobic core is formed at the interface between the A-face and a long α-helix. 

The crystal structure shows that D55, whose original residue is N55 modified with a glycan, is located at the C-terminal end of the long α-helix and is far distant from the B-face containing the IBS. The TH activity of N55D is identical to that of the wild-type, confirming the previous indications that the glycan is not involved in ice binding [25]. The C-terminal region (A204–V216) is longer than that of other microbial IBPs. In the crystal structure of *Anp*IBP, the C-terminal region was not identified because of the weak electron density, suggesting that it is flexible. Similar additional segments at the N- or C-terminal end have sometimes been identified in the DUF3494 microbial IBPs, though their functions are not clear. The C-terminal segment of *Anp*IBP contains basic residues (K206, R209, K212, and K215) at equal intervals, which sandwich several hydrophobic residues. The role of this C-terminal segment is unknown. The overall structural motif of *Anp*IBP is highly similar to that of other IBPs from bacteria, despite their phylogenetic distance. This finding is in good agreement with the previous indications that Antarctic ascomycetes acquired *Anp*IBP from bacteria through horizontal gene transfer [25]. 

### 3.3. The IBS of AnpIBP Located on the B-Face

To identify the IBS location of *Anp*IBP, we prepared a mutant protein in which the T58 on the A-face (Figure 3A) was replaced with tyrosine (denoted T58Y) and examined its TH activity. The S153 and T156 on the B-face were also replaced with tyrosine, and the TH value was measured. The Y131 on the B-face was further replaced with threonine. Figure 3B shows the TH activity of *Anp*IBP (wild-type) and its four mutant proteins: T58Y, Y131T, S153Y, and T156Y. They showed a typical linear dependence of TH on the square root of concentration. The wild-type had 0.63 °C of maximal TH activity at a concentration of 250 µM. The T58Y that mutated on the A-face exhibited a similar dependence to the wild-type. Mutations in S153Y and T156Y on the B-face resulted in a decrease in activity to 17% of the wild-type. These results suggest that the IBS of *Anp*IBP is on the B-face. The Tyr131 located on the B-face protrudes a bulky side chain, which may cause steric hindrance when the B-face binds to an ice crystal. Hence, we expected TH to be increased for Y131T, because threonine is a typical ice-binding residue for most IBPs. However, the mutation affected neither TH nor the ice-shaping abilities, indicating that Y131 is located in such a way as to maintain the ice-binding ability. To confirm that there was no significant influence of these mutations on protein folding, we performed circular dichroism (CD) measurements (Appendix A). The wild-type (*Anp*IBP) exhibited minimum ellipticity at 217 nm and maximum ellipticity at 197 nm on the CD spectra, suggesting a β-helical structure. The same profile was reported for other microbial IBPs [30,42]. All *Anp*IBP mutants exhibited highly similar CD profiles to the wild-type, suggesting that defective mutations did not affect the overall structure but collapsed a structure needed for ice binding on the B-face. 

To further investigate the ice-binding properties of the defective mutants, we observed their ice-shaping ability and FIPA pattern (Figure 3C,D). It has been demonstrated that the morphology of a single ice crystal showed a lemon shape in hyperactive IBPs exhibiting 5–6 °C of TH activity, such as insect-derived IBPs (Figure 3Ca). In contrast, a hexagonal bipyramid is created in moderately active IBPs exhibiting approximately 1 °C of maximal TH, such as fish-derived IBPs (Figure 3Cb). *Anp*IBP is categorized as a moderately active IBP based on its TH value (Figure 3B). However, a single ice crystal in *Anp*IBP solution represented a lemon-like shape (Figure 3Cc) as observed previously [25], suggesting that *Anp*IBP can also bind to multiple ice planes although its manner will not be perfectly the same as the hyperactive IBPs. In contrast, the defective mutant S153Y generated a hexagonal bipyramid, as produced by fish IBPs (Figure 3Cd). Another mutant, T156Y, also produced a hexagonal bipyramid, although it is not clearly faceted (Figure 3Ce). In FIPA analysis, the illumination of S153Y was observed in six patches at the middle latitude of the ice hemisphere (Figure 3Da). This result indicates that the S153Y mutant loses the binding affinity to the prism ring, as it occupies the equator. The illumination area of T156Y was further reduced (Figure 3Db) in comparison with that of the S153Y mutant, suggesting a significant loss of binding ability. These results suggest that Ser153 and Thr156 on the B-face are the residues necessary to bind to the water molecules constructing the prism ring of a single ice crystal hemisphere. 

### 3.4. Ice-Like Polygonal Water Networks Were Observed on the IBS of AnpIBP

The data presented here indicate that the IBS of *Anp*IBP is constructed on the B-face, similar to other microbial IBPs [29]. On the basis of the known IBS construction, its location in *Anp*IBP was hypothesized to be in the area surrounded by a blue line in Figure 4A. The size of this IBS—the solvent-accessible area—is approximately 1,418 Å^2^, slightly larger than the 921 Å^2^ estimated for IBS on *Tis*IBP [26]. The amino acid residues that participate in the IBS are also shown in Figure 4A. The Thr and Ser are indicated in yellow, and the charged residues are in green. A four-residue segment stacked on the β1-loop and six-residue segments of six-stranded parallel β-sheets are the candidate ice-binding residues: T36-S-I-T39 (loop region), Q17-G-V-A-N-A22 (β1), S189-A-V-T-L-D194 (β6), A171-L-I-A-A-Q176 (β5), S153-S-A-T-L-G158 (β4), T126-T-L-V-T-Y131 (β3), and S100-A-V-G-L-D105 (β2). For each β-loop, the side chains of four residues are directed outward, whereas two residues are directed inward, leading to an out–out–in–out–in–out pattern similar to that identified in *Tis*IBP6 and *Col*IBP [26,30].

Some of the DUF3494 IBPs are composed of repetitive amino acid sequences. For example, *Ff*IBP, a hyperactive IBP, has repetitions of the consensus sequence –T–A/G–X–T/N [44]. A slightly different *Efc*IBP also contains the repetitive sequence –T–X–T or T–X–D and is categorized as a moderately active IBP [45]. In the IBS amino acid sequence of *Anp*IBP, there is no such repetitive sequence. The outward-pointing residues on the IBS of *Anp*IBP contain relatively small hydrophobic side chains, such as alanine, glycine, serine, and threonine, which share 70% of the total IBS. The residues shown in columns 1, 2, and 4 (Figure 4A, right) are hydrophobic residues, whereas column 6, at the edge, contains charged groups. A hydrophobic surface containing spaced hydroxyl groups is a typical feature of IBS and is thought to be key to locating the ice-like clathrate water molecules on that site.

It has been shown that hydration water molecules on an IBP are relevant to its ice-binding function. We examined such water molecules located on the crystal structures of *Anp*IBP and the S153Y mutants. In crystals of *Anp*IBP and S153Y, the asymmetric unit contained six molecules and three molecules, respectively (Appendix A). Among the six *Anp*IBPs, half of the IBSs were covered by another protein, whereas the other half was exposed to the solvent. This is due to the hydrophobic nature of IBSs, which tend to align face to face in an asymmetric unit [47]. We found that approximately 150 water molecules are located beside three molecules of *Anp*IBP that are closely assembled in the crystal. These water molecules mostly form pentagonal or hexagonal structures. These structures are called “semi-clathrate water networks” and are initially found in a hyperactive fish type I IBP called Maxi [22,23]. In Maxi, approximately 400 water molecules participate in the water networks, which extend from a core region to the outside of the four-helix bundle structure. When we identified the water molecules whose oxygen atoms are proximal to each other within 3.5 Å to be connected with a hydrogen bond [48,49], a total of 88 water molecules were displayed on the solvent-exposed IBS (Figure 4B). The molecules are mainly organized into polygonal arrangements similarly to those observed in Maxi, which extend over the left and right sides of the IBS in *Anp*IBP. There are at least 13 polygons on the IBS of *Anp*IBP, which are mainly located around S153 and T156. Such ice-like polygonally arranged water molecules were not identified on the solvent-exposed IBS of a mutant, S153Y (Figure 4C), whose TH activity and ice-binding ability were lowered (Figure 3C,D). Molecular dynamics calculations of ice growth kinetics suggest that the growth front of an ice crystal forms a unique layer composed of semi-clathrate waters that change between liquid and solid intermediate states [50,51]. This growth front is named a “quasi-liquid layer” and is assumed to be an initial target of IBP when making an IBP–ice complex. The polygonal water networks of *Anp*IBP are assumed to merge with this quasi-liquid layer, leading to the location of the host (*Anp*IBP) on the surface of an ice crystal. This mechanism facilitates rapid association of IBP to ice crystals and has been proposed for several other IBPs that accompany the polygonal water networks. 

### 3.5. Ice-Binding Mechanism of AnpIBP

Figure 5A,B shows a single ice crystal hemisphere formed in a solution and its constituent water molecules organized into hexagonal arrangement, respectively. Figure 5A illustrates the location of the prism ring as observed for *Anp*IBP, as well as the basal plane. The basal plane is the polar area of the spherical ice corresponding to the top surface of the hexagonal ice (Figure 5B). The location of the primary and secondary prism planes (white squares) and the oxygen atom spacing of the basal and primary prism planes (expanded views) are indicated in Figure 5B.

It has been shown that several IBPs have regular arrays of hollows on their surface [5]. They are sometimes vacant in the crystal structure due to crystal packing. They are thought to trap the hydration water molecules so that they are located at regular intervals [8,9]. For such arrayed water molecules, an anchoring role to bind the host protein (IBP) to ice crystal plane(s) has been hypothesized. For example, an insect-derived *Tm*IBP has six arrayed hollows on its IBS to trap water molecules (Figure 5C). This IBP consists of tandem repeats of 12 consensus sequences containing a tripeptide “TxT”, where T is Thr and x is any amino acid residue. In this motif, two ranks of six Thr residues are exposed outside and x is oriented inside. This out–in–out arrangement of TxT is stacked on the β-helical domain, leading to the formation of hollows in which to locate six regularly arrayed water molecules (Figure 5C). An original ice-binding model of *Tm*IBP was that the two ranks of Thr function in ice binding, because the two-dimensional spacing between their side-chain hydroxyl groups shows complementary matches to the oxygen atom spacings of the primary prism and basal planes [9]. *Tm*IBP, however, binds to all of the ice planes besides the primary prism and basal planes. Figure 5C shows that the hollows are present at 4.6- and 13.8-Å intervals, closely matching the oxygen atom spacings of the basal plane (Figure 5B). This observation is consistent with *Tm*IBP binding to the basal plane. Because the oxygen atom distance of 4.6 ± 1.0 Å has been identified for all planes of a single ice crystal, the presence of surface-trapped oxygen atoms separated by 4.6 Å might be key to binding to the oxygen atoms constructing many ice planes. 

The presence of four regularly arrayed hollows was identified in the IBS of *Anp*IBP (Figure 5D). Unlike *Tm*IBP, these hollows are not flanked by ranks of Thr, as the IBS of *Anp*IBP does not consist of a tandem repeat sequence. The segments organized into out–out–in–out–in–out arrangement stacked on the β-helical domain create these hollows with regular intervals. The distance between the first and second hollows and between the third and fourth hollows is 4.6 Å, and that between the first and third hollows and between the second and fourth hollows is 14.7 Å (Figure 5D). These intervals perfectly match the oxygen atom distance in the primary and secondary prism planes (4.6 and 14.7 Å) as well as that comprising the prism ring. The 4.6 Å atom spacing is common for many ice planes, whereas 14.7 Å is only specific to the prism planes. We assume that such a rigorous water atom spacing is located on the prism planes but not on the basal plane. Figure 5E shows an additional surface model of a microbial IBP denoted *Tis*IBP6 [26]. This is also composed of a β-helical domain that includes the out–out–in–out–in–out segments and exhibits the ability to bind to whole ice crystal planes. The hollows on this IBP are separated by 4.6 Å, and the 14.7 Å interval does not exist between these water molecules. 

The *Anp*IBP has the polygonally arranged ice-like water molecules on its IBS (Figure 4B), which are assumed to have an anchoring role by merging with the quasi-liquid layer constructing an ice crystal surface, which is more ordered than bulk water molecules, but less ordered than those in the hexagonal ice lattice [52,53]. The polygonally arranged water molecules are assumed to assist the *Anp*IBP in immersing into the quasi-liquid layer, which may locate the IBS in close proximity to the water molecules of the solid ice lattice. This structure facilitates subsequent *Anp*IBP binding to the latticed water molecules separated by 14.7 Å that are specifically involved in the prism ring. A similar, but not identical arrangement of water molecules was observed on a fish type III IBP (*nfe*IBP6 and its mutant) [21]. This IBP also has polygonally arranged water molecules but does not exhibit hollows on that surface. Instead, several water molecules that participate in the polygonal water network exhibit a perfect position match to their target ice planes (primary and pyramidal ice planes). The insect-derived IBPs, such as *Tm*IBP, have regularly arrayed hollows, but they do not accompany a polygonally arranged water network [9]. The other IBPs that have polygonal water networks do not have arrayed hollows on their surface [4,20]. Therefore, *Anp*IBP is the first example to both have polygonally arranged water molecules and have hollows to tightly trap the surface water molecules at regular intervals, allowing the binding of this IBP only to water molecules separated by 14.7 Å, which form the prism ring. The fish type III IBP also accompanies the surface-bound waters [21], which are not linearly aligned but are two-dimensionally organized. This may cause its specific binding to only the primary prism plane and a pyramidal plane. The growth speed of the prism planes is approximately 100 times faster than that of the basal plane [12]. *Anp*IBP binding to the prism planes effectively inhibits this fast growth, which protects the host organism (*A. psychrotrophicus*) from ice injuries.

To summarize, the present study determined the target ice planes and X-ray crystal structures of the wild-type and a defective mutant of *Anp*IBP. The *Anp*IBP uniquely binds to the water molecules that participate in the primary and secondary prism planes, which were detected as a prism ring in the FIPA experiment. Both the ice-like polygonally arranged water molecules and a row of four hollows to trap the surface water molecules exist on the IBS of *Anp*IBP and perfectly match the oxygen atom spacings (4.6 and 14.7 Å) of the prism ring. The presence of oxygen atoms separated by 14.7 Å should be key to recognizing the prism ring, whereas four trapped water molecules will bind *Anp*IBP tightly onto the prism ring. These results suggest that *Anp*IBP is fine-tuned to merge with the ice–water interface of an ice crystal through polygonally arranged water molecules and then bind to specific water molecules constructing the prism ring through the water molecules trapped in the four hollows. 

## Figures and Tables

**Figure 1 biomolecules-10-00759-f001:**
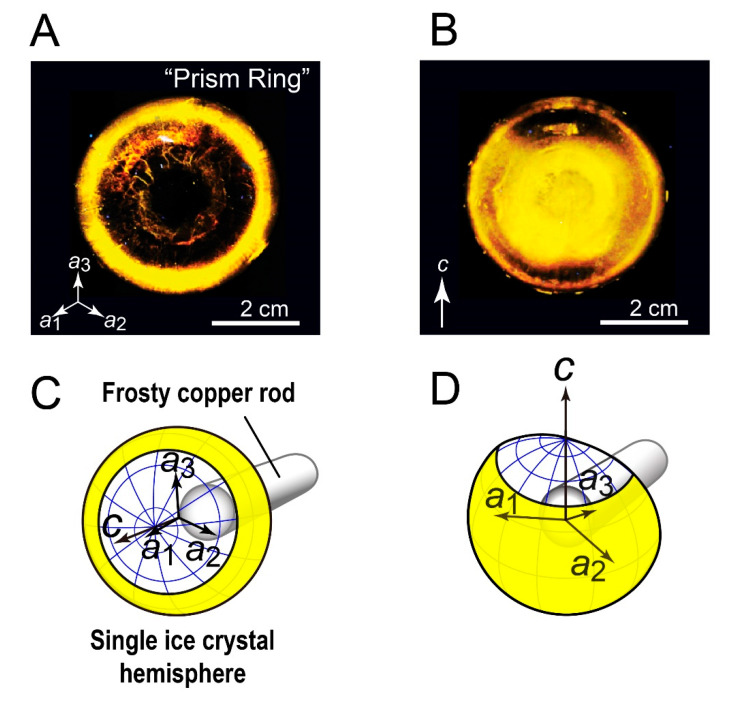
Observation of fluorescence-based ice plane affinity (FIPA) of *Anp*IBP. (**A**) A ring-like FIPA pattern of *Anp*IBP (denoted a “prism ring”). The ice hemisphere was mounted on a frosty copper rod to align its *c*-axis with the rod as illustrated in (**C**), and then soaked in 0.1 mg/mL protein solution to direct the *c*-axis downward. The prism ring was observed at the rim of the hemisphere, suggesting that *Anp*IBP binds to the water molecules that participate in both the primary prism and secondary prism planes. (**B**) The FIPA pattern of *Anp*IBP observed on a different ice hemisphere whose *a*_3_-axis is normal to the paper as illustrated in (**D**). The pattern shows that the prism ring observed in (**A**) is spread from the equator to a higher latitude.

**Figure 2 biomolecules-10-00759-f002:**
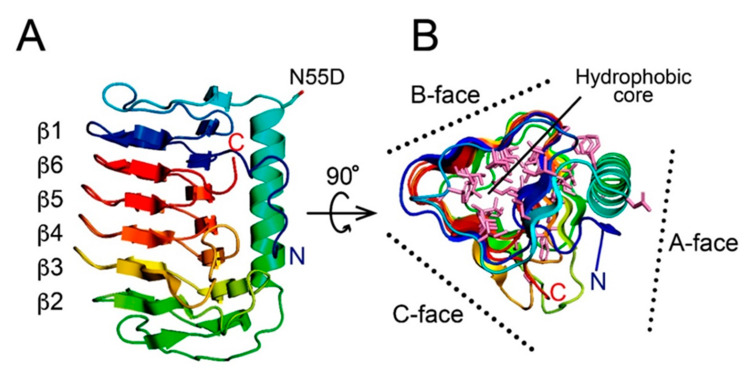
The crystal structure of *Anp*IBP (7BWX.pdb). (**A**) Overall structure of *Anp*IBP represented with ribbon model colored in gradation from blue (N-terminus) to red (C-terminus). The six β-helical loops are irregularly aligned in the order β1–β6–β5–β4–β3–β2. (**B**) Structural view of *Anp*IBP along the β-helical axis, showing that the β-helical fold is a triangular shape on which the A-, B-, and C-faces are defined. A large hydrophobic region is present at the center of the triangle. The long α-helix is stabilized on the A-face through additional hydrophobic cores.

**Figure 3 biomolecules-10-00759-f003:**
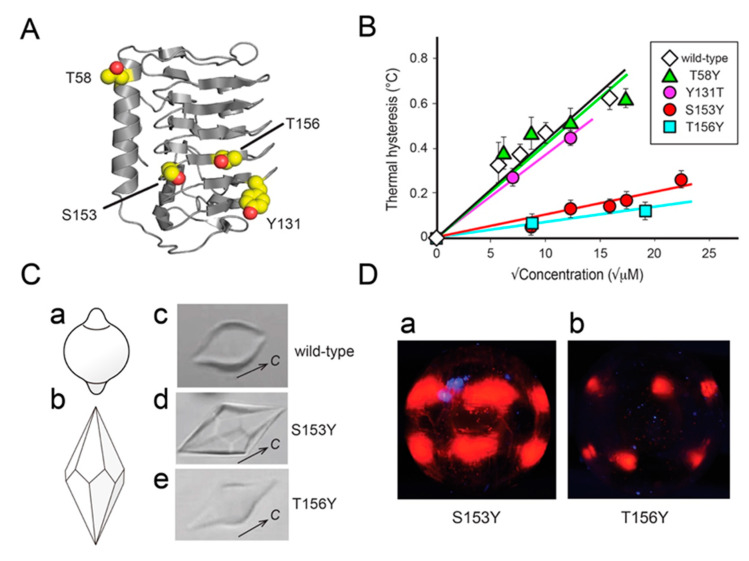
Effect of amino acid replacement on the ice-binding ability of *Anp*IBP. (**A**) Mutation sites on the crystal structure of *Anp*IBP. The side chains of the wild-type residues are shown as spheres. T58 is located on the A-face and the others on the B-face. (**B**) Thermal hysteresis activities (°C) of *Anp*IBP and its mutants plotted against the square root of each concentration. (**C**) Morphologies of a single ice crystal observed for *Anp*IBP and its mutant solutions. Illustrations show a lemon-shape ice crystal (a) observed for hyperactive IBPs and a bipyramidal ice crystal (b) observed for moderately active IBPs. The wild-type *Anp*IBP showed a lemon-shaped ice crystal (c), whereas the defective mutants (S153Y and T156Y) showed crystal bipyramids (d and e). (**D**) Comparison of FIPA patterns between the S153Y and T156Y mutants of *Anp*IBP. The orientation of these ice crystal hemispheres are the same as in Figure 1B,D. These mutants lost the ability to bind to the prism ring.

**Figure 4 biomolecules-10-00759-f004:**
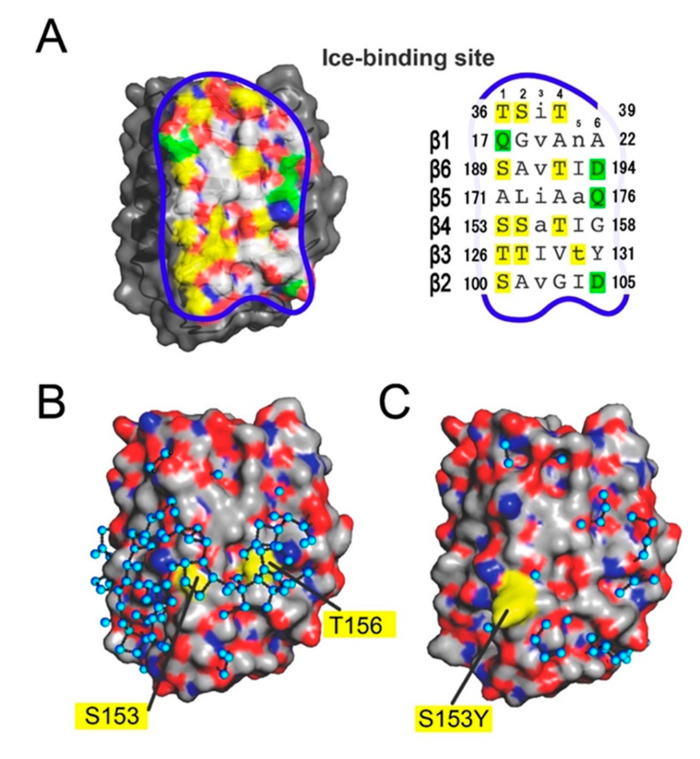
Surface models of the present determined structures of *Anp*IBP (7BWX.pdb) and its mutant (7BWY.pdb). (**A**) An ice-binding site (IBS) constructed on the B-face of *Anp*IBP (wild-type) whose area is indicated by a blue line. The capitals in columns 1, 2, 4, and 6 represent the outward-pointing residues, mostly Thr and Ser (yellow). The small characters in columns 3 and 5 are inward-pointing residues, mostly hydrophobic residues. The polar residues are aligned in column 6 (green). (**B**) Surface model of wild-type *Anp*IBP in the same orientation as (**A**), where gray, red, and blue show the hydrophobic, positively charged, and negatively charged residues, respectively. The black rods connect the oxygen atoms of the hydration waters (cyan) proximal to each other within 3.5 Å, the maximum allowed distance for a hydrogen bond [47,48]. Many hydration water molecules are circled and organized into polygonal geometry on the IBS. (**C**) Surface model of the crystal structure of the S153Y mutant whose ice-binding ability was lowered. The polygonally arranged water molecules were not detected on the IBS.

**Figure 5 biomolecules-10-00759-f005:**
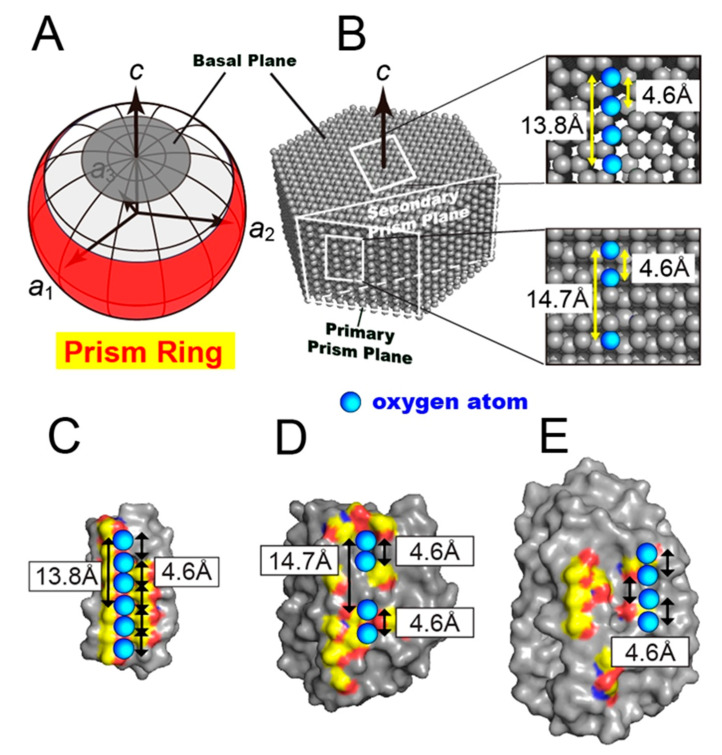
Oxygen atom spacings of a single ice crystal and IBPs. (**A**) Illustration of the prism ring, indicating the *Anp*IBP-bound area on a single ice crystal hemisphere. (**B**) Hexagonally arranged water molecules that are the constituents of the single ice crystal hemisphere. Only the oxygen atoms are only indicated. The basal plane is the top surface of (**A**,**B**). The oxygen atoms separated by 4.6 and 14.7 Å are unique constituents of the primary and secondary prism planes as well as the prism ring. (**C**) IBS of *Tm*IBP has six hollows to trap the water molecules (cyan) at regular intervals. They show a close match to the O-atom distances of the basal plane (4.6 and 13.8 Å). (**D**) Four linearly arrayed hollows observed on the IBS of *Anp*IBP to trap the water molecules. Their intervals perfectly match the distance between the oxygen atoms comprising the prism ring (4.6 and 14.7 Å). (**E**) Hollows with 4.6-Å intervals observed on *Tis*IBP6, which possesses the ability to bind to multiple ice planes.

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
