# Peer review of "An Ice-Binding Protein from an Antarctic Ascomycete Is Fine-Tuned to Bind to Specific Water Molecules Located in the Ice Prism Planes"

_biomolecules, 2020, doi:10.3390/biom10050759_

Round 1

Reviewer 1 Report

The manuscript presents the information on the structure and ice binding properties of a new antifreeze protein extracted from fungus. The results are new and can be interesting for researchers studying antifreeze proteins.

In general, the paper is well-written and gives good impression. However, some issues cause questions. Figure 1a shows ice plane affinity of the protein molecules visualized by fluorescence technique. The authors discuss accumulation of the protein mainly near an equator of the ice crystal hemisphere (large yellow ring in Figure 1a). But the hemisphere also has a small yellow ring at the pole. Perhaps, the protein adsorbs not only at the prismatic faces of ice crystals, but also at the faces having more complicated orientation (these faces are not pyramidal faces). The authors are suggested to provide in the manuscript more detailed discussion on this matter.

Reviewer 2 Report

The manuscript is a thorough examination of the AnIBP antifreeze mechanism, and is very well written. The most important issue is the different crystal forms of the wild-type and mutated AnIBP (Line 178 - 180). Can the authors confirm that potentially different crystal packings did not affect their interpretation of the data? This is especially important for Figure 4 and interpretation of the protein's ice-binding mechanisms. Comparisons are made between these two proteins and there is a pressing need to ensure that the different arrangement of waters isn't due solely to crystal packing differences. The authors do discuss some of this in the Discussion section, but I would prefer to see a supplemental figure showing the packing arrangement.

The remaining comments are relatively minor in nature and can be easily addressed.

Line 42  - 45. I would suggest moving this 'concluding sentence' to the end of the Introduction section. It is rather disturbing located here.

Line 102 - 103. Not all insect AFPs have this consensus sequence. Please modify the text accordingly.

Line 116: "lower TH activity" Unfortunately English is ambiguous with negative numbers. Do the authors mean poorer or better activity relative to fish?

Line 194: "Unique 'prism-ring'". I would prefer the term 'novel', because we do not know yet if this prism-ring binding occurs among other IBPs.

Figure 1. I understand the need to include type III IBP FIPA, but I find it confusing to start with that in panels A and B. I would prefer first presenting the AnIBP data, then mentioning the type III AFP data. It would also be helpful to clear indicate in the panels which protein is being tested.

Line 227 - 230: Include "data not shown" at the end of the sentence

Line 248: "present determined" can be deleted.

Reviewer 3 Report

This study is a well designed investigation of an IBP from a fungi, anpIBP, which behaves like a moderate IBP. The characterization of the protein is rigorous and convincing. The claim that water molecules on the IBS are organized in a way that matches some of the ice crystal planes was demonstrated here in a convincing manner. I have only a few comments:

  1. The finding of the prism ring is interesting, but the authors did not explain why for IBPIII, no ring was obtained. Does anpIBP bind to the prism and secondary prism planes and that leads to the prism ring? In other words, binding to which crystal planes creates the prism ring? This point should be explained better.
  2. The crystal morphology obtained for the wild type (fig. 3C) is not a lemon shape such as the morphology of ice in TmAFP solutions. The morphology obtained in anpIBP is very similar to that of the large AFGPs (AFGP1-5), where clear sharp tips are observed. It would be better not to confuse the readers with these definitions.
